# The Role of the Small Bowel in Unintentional Weight Loss after Treatment of Upper Gastrointestinal Cancers

**DOI:** 10.3390/jcm8070942

**Published:** 2019-06-28

**Authors:** Babak Dehestani, Carel W le Roux

**Affiliations:** Diabetes Complications Research Centre, University College Dublin, D04 HV52 Dublin, Ireland

**Keywords:** gastrointestinal, cancer, small bowl, unintentional, weight loss

## Abstract

Upper gastrointestinal (GI) cancers are responsible for significant mortality and morbidity worldwide. To date, most of the studies focused on the treatments’ efficacy and post-treatment survival rate. As treatments improve, more patients survive long term, and thus the accompanying complications including unintentional weight loss are becoming more important. Unintentional weight loss is defined as >5% of body weight loss within 6–12 months. Malignancies, particularly GI cancers, are diagnosed in approximately 25% of patients who present with unintentional weight loss. Whereas some recent studies discuss pathophysiological mechanisms and new promising therapies of cancer cachexia, there is a lack of studies regarding the underlying mechanism of unintentional weight loss in patients who are tumor free and where cancer cachexia has been excluded. The small bowel is a central hub in metabolic regulation, energy homeostasis, and body weight control throughout the microbiota-gut-brain axis. In this narrative review article, the authors discussed the impacts of upper GI cancers’ treatment modalities on the small bowel which may lead to unintentional weight loss and some new promising therapeutic agents to treat unintentional weight loss in long term survivors after upper GI operations with curative intent.

## 1. Introduction

Upper gastrointestinal (GI) cancers include oesophageal, stomach, duodenal, liver, gallbladder, biliary tract, ampulla of Vater, and pancreatic cancers. These cancers are responsible for significant mortality and morbidity worldwide [1]. According to the GLOBOCAN 2018, cancers of the stomach, liver and esophagus together represent 13.5% of all cancers in 2018, but are responsible for 26.2% of the mortality due to cancer [1].

The GI tract is an integrated system with signaling between organs, tissues and cells. Unsurprisingly, the most common site of distant metastasis of upper GI cancers is the liver [2,3,4,5,6]. Treatment options for upper GI cancers include surgery, chemotherapy, radiation therapy, chemoradiation therapy, targeted therapy, and surgical/endoscopic palliative interventions [7,8,9,10]. Consequently, defining a therapeutic plan for each patient requires a multidisciplinary approach including gastroenterologists, pathologists, GI surgeons, oncologists, radiologists, radiation oncologists, general practitioners, dietitians, psychiatrists/psychologists and social workers [7,11,12]. 

To date, most of the studies focused on the efficacy and post-treatment survival rate. As treatments improve, more patients survive long term, and thus understanding altered gut-brain signaling pathway and unintentional weight loss are becoming more important.

Unintentional weight loss is defined as >5% of body weight loss within 6–12 months. Malignancies, particularly GI cancers, are diagnosed in approximately 25% of patients who present with unintentional weight loss [13,14,15]. Moreover, the highest frequency of weight loss is seen among the patients with pancreatic or gastric cancer [16,17]. Although the cancer in itself can cause weight loss, the treatment which may include surgical rearrangement of the small intestine may also contribute. If unintentional weight loss after surgery with curative intent is between 10 and 30% from starting weight, then long term morbidity and mortality increase even in those in remission of cancer [18,19,20,21,22].

Unintentional weight loss can be considered as both a sign and a complication of cancers and their treatment. Some recent studies discussed pathophysiological mechanisms and new promising therapies for cancer cachexia which is characterized by the ongoing loss of skeletal muscle mass with or without loss of fat mass [17,23,24,25,26]. However, there is a lack of studies regarding the underlying mechanism of unintentional weight loss in patients who are tumor free and where cancer cachexia has been excluded.

## 2. Role of the Small Intestine in the Gut-Brain Axis

The cycle of repetitive episodes of hunger and satiation depends on harmonized bidirectional communication between the subcortical areas of the central nervous system (CNS) and GI system via endocrine, neurocrine and paracrine interactions [27]. The gut-brain axis (GBA) includes the central nervous system (CNS), both brain and spinal cord, the autonomic nervous system (ANS), the enteric nervous system (ENS) and the hypothalamic pituitary adrenal (HPA) axis. Gut-microbiota have also emerged as having important roles in the microbiota-gut-brain axis [28,29,30,31].

Satiety gut hormones, such as glucagon-like-peptide 1 (GLP-1), Peptide YY (PYY) and oxyntomodulin (OXM) are released from L-cells when food comes into contact with the small bowel mucosa [32]. L-cells are present from duodenum to the rectum, however, they are most abundant in the ileum and colon. The early response of duodenal L-cells in releasing satiety hormones may be enhanced by neural regulation and intraluminal short-chain fatty acids (SCFAs), bile acids and amino acids [32,33,34,35].

Bile acids are now considered major signaling molecules as they activate Farnesoid X receptor (FXR) and G protein-coupled bile acid receptor 5 (TGR5). This leads to stimulating secretion of GLP-1 and regulates glucose and energy homeostasis, lipid metabolism, and gut microbiota [36,37,38,39]. Visceral signaling of satiation is further enhanced by small bowel gut microbiota. Short-chain fatty acids (SCFAs) including propionate, butyrate and acetate which are mainly produced via fermentation of dietary fibers by gut microbiota, also increases G-protein mediated secretion of PYY and GLP-1 [39,40,41]. The small bowel is thus a central hub in metabolic regulation, energy homeostasis, and body weight control.

## 3. Impacts of Upper GI Cancers’ Treatment Modalities on the Small Bowel

### 3.1. Surgical Procedures

Operations of the upper GI Cancers including esophagectomy, gastrectomy, Whipple procedure and extended (radical) cholecystectomy involve re-routing the normal GI tract. In particular, the anatomy of the gastroduodenal junction and proximal small intestine are altered. Hence, after these operations, foods rapidly progress into and through the small bowel [42,43,44]. The mentioned prompt contact of food causes small bowel mucosa adaptation [45,46,47] and consequently, an increase in absolute number of enteroendocrine L-cells, particularly in the proximal small bowel which finally leads to exaggerated release of gut satiety hormones [48,49]. After esophagectomy at 6 weeks and 3 months, body weight decreases and postprandial release of satiety gut hormones (GLP-1) are increased [48]. Another study confirmed that patients after esophagectomy had significant body weight loss at 3, 6, 12, and 24 months which was associated with increases in postprandial GLP-1 and PYY responses in comparison with the controls [49]. 

Simultaneously, excess bile enters the small bowel resulting in elevated plasma bile signals [50]. Furthermore, the gut microbiota profile changes after the surgeries which lead to enhancing visceral signals of satiation to the brain reward centers [51,52,53]. 

In contrast to the satiety gut hormones, Ghrelin as a hunger hormone is mostly secreted by X/A like cells of the fundus of the stomach [54,55], hence, its secretion is initially reduced after curative upper GI cancer surgery. Aside from ghrelin’s stimulatory effect on releasing growth hormone (GH) [56,57], it has profound orexigenic and adipogenic characteristics via activating the arcuate nucleus containing neuropeptide Y (NPY) and agouti-related peptide (AgRP), and inhibiting neurons containing proopiomelanocortin (POMC) in hypothalamus which finally induces feelings of hunger [58,59]. Even though initial ghrelin reduction after upper GI cancer surgery may lead to unintentional weight loss, it cannot fully explain long term unintentional weight loss since ghrelin recovers to baseline levels after 12–24 months [60,61]. Ghrelin analogues have also had limited success in increasing long term food intake and bodyweight [60,61].

Harris et al. showed that loss of taste (hypogeusia) and smell (hyposmia) occurred at nearly 50% of patients after upper GI cancer surgery. Although it resolves in most patients within 6–12 months, it might impact on changing food preferences, contributing to unintentional weight loss [62]. Taken together, the reduced energy intake and unintentional weight loss after upper GI cancer surgery may be partly explained by the alterations in the gut-brain axis including gut mucosal adaptation; increased visceral signals to the brain; reduced brain reward responses to food; reduced eating behavior; reduced food intake and changed food preferences.

### 3.2. Chemotherapy and Radiotherapy

Chemotherapy (cytotoxic drugs) also target healthy fast-dividing cells including GI tract cells, particularly proliferating enterocytes in the small bowel and colon which can lead to GI mucositis. Consequently, barrier dysfunction and epithelium impairment occurred in GI mucositis [63,64,65]. Likewise, abdominopelvic radiotherapy adversely affects the normal enterocytes which leads to radiation-induced GI mucositis [66,67,68]. A wide range of GI symptoms in cancer patients including diarrhea, nausea, vomiting, abdominal pain, malnutrition, bleeding, fatigue and unintentional weight loss may be a consequence of mucositis [65,67]. 

Gut-microbiota play a significant role in the pathogenesis of mucositis of the GI tract. Montassier et al. found that abundances of Firmicutes and actinobacteria were markedly decreased while there was a considerable increase in abundances of proteobacteria in fecal samples of patients after chemotherapy compared to the samples collected before chemotherapy [65]. The phyla Firmicutes also decreased by 10% whereas phyla Fusobacterium increased by 3% after radiation therapy [66]. Bajic et al. suggested chemotherapy causes an integrated dysregulation through the microbiota-gut-brain axis since the most vulnerable cells to the toxic side effects of cytotoxic drugs are GI tract cells, CNS cells and the gut microbiota [64]. However, this does not fully explain the underlying reasons of ongoing unintentional weight loss after chemo/radio therapy-related complications have resolved.

## 4. New Promising Therapeutic Agents

Octreotide, a synthetic somatostatin analogue, acts via somatostatin receptors (SSTRs), a family of G-protein coupled membrane receptors [69,70], to inhibit the satiety gut hormone responses. This may hypothetically lead to stimulation of the brain reward center and increases in food intake and body weight. The inhibition of the satiety gut hormone responses with a somatostatin analogue almost doubled ad libitum food intake [49]. Elliott et al. showed that administration of octreotide in patients after esophagectomy led to attenuation of exaggerated postprandial satiety gut hormone responses which was associated with increased appetitive behavior toward a sweet-fat stimulus [71].

Moreover, patients with progressive cancers or severe dumping syndrome after gastrectomy are often treated for prolonged periods with chronic octreotide [72]. Therefore, octreotide might be considered as a potential therapeutic agent for unintentional weight loss after upper GI cancer surgery. The drug has a well-documented safety profile, is frequently used in patients with cancer and is known to be well-tolerated [72,73,74]. However, the main side effect of chronic octreotide includes gallstone formation that may result in cholecystitis and pancreatitis [75,76].

Ghrelin receptor agonists such as Capromorelin and BIM-28163 can increase appetite and food intake which causes weight gain. The administration of Anamorelin—a non-peptide, orally active, selective agonist of the ghrelin/GH secretagogue receptor—demonstrated some clinical benefits in terms of appetite stimulation, improving lean body mass and weight gain in patients with cancer cachexia [77,78,79,80,81,82]. Therefore, these novel drugs might be some potential therapeutic agents for the patients after treatment with curative intent who are suffering from ongoing unintentional weight loss [83,84].

Decreasing the chances of weight reduction initially right after the cancer surgery or during chemo/radiation therapy may have some beneficial impacts in the long term for these patients. Suitable nutritional support by dietitians postoperatively and considering alternative ways of feeding including parenteral nutrition (PN) or feeding jejunostomy in indicated patients may also attenuate initial weight loss [82,85,86]. Probiotics may also have therapeutic effects on chemo/radiation therapy induced GI mucositis by restoring the normal composition of the gut microbiota [67,87].

GABA (γ-aminobutyric acid) is the major inhibitory neurotransmitter of the CNS, and NPY/AgRP neurons release GABA to directly inhibit POMC neurons in the hypothalamic arcuate nucleus (ARC) which may result in increased food intake and weight gain [81,88,89]. Hence, GABA receptor agonists and GABA reuptake inhibitors might be considered as potential orexigenic agents.

Leptin is an anorexigenic hormone which is secreted by adipocytes and regulates energy expenditure, appetite and body weight. Some animal studies indicate that the administration of leptin receptor antagonists leads to significant increases in food intake and body weight in rodents [90,91,92]. Hence, these antagonists might be potential therapeutic options for unintentional weight loss.

Alpha-melanocyte-stimulating hormone (α-MSH) is a main product of POMC neurons which exerts its anorexigenic effects via the melanocortin 3 (MC3) and 4 (MC4) receptors in the hypothalamic paraventricular nucleus. Thus, MC4 antagonists such as BL-6020/979 may ameliorate unintentional weight loss and loss of lean body mass in patients after cancer surgeries [93,94,95].

Fecal microbiota transplantation (FMT) is mainly used for treating recurrent *Clostridium difficile* infection, inflammatory bowel disease (IBD) and functional gastrointestinal disorders (FGID) [96,97,98,99]. However, recent studies demonstrated that FMT may result in altering energy homeostasis, appetite, food preference and body weight due to the new composition of recipient’s gut-microbiota after transplantation which is changed by the stool donor’s gut microbiota [100,101]. A case study showed that a 32-year-old female developed new-onset obesity after receiving stool from a healthy but overweight donor. Her BMI increased from 26 before FMT to 34.5 at 36 months post-FMT [102]. Animal studies also support transmissible obesity via FMT in mice [103]. Therefore, performing FMT from healthy overweight stool donors may lead to increased food intake and weight gain in patients with unintentional weight loss after cancer treatment.

Treatments for upper GI cancers have improved significantly and more patients are surviving long term. Long term unintentional weight loss may increase morbidity and mortality while also reducing quality of life. The microbiota-gut-brain axis may hold the key to understand and treat unintentional weight loss in long term survivors after upper GI operations with curative intent.

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
