# Peer review of "The Role of the Small Bowel in Unintentional Weight Loss after Treatment of Upper Gastrointestinal Cancers"

_jcm, 2019, doi:10.3390/jcm8070942_

Reviewer 1 Report

This is a nice narrative review, giving many references and thereby opening many questions and stimulating discussion.

It is indeed a special  manuscript, neither original work nor pure review. The difference to a  standard review article is the fact that data are interpreted and used  for ideas, hypotheses and speculations. 

Format: ok, with adequate subtitles.

Statement:  stimulating hypotheses and associations between regulations of appetite and post-surgical anatomy

Method and Material: broad lit. review

Conclusion (line 183-184): this  is somehow incomplete: I agree the gut-brain axis is important but the  altered gut microbiota also plays a key role and should also be  mentioned in the conclusion.

Author Response

Dear Reviewer, 

Thanks a million for your invaluable comments. Indeed, we are really pleased that you've found our article as a nice narrative review. 

I totally agree with your comment regarding the conclusion (line 183-184), hence I revised it as requested to highlight the pivotal role of altered gut microbiota. 

Best wishes, 

Reviewer 2 Report

This review provides a well-written summary and overview of upper GI cancer-associated unintentional weight loss, with an interesting perspective of focusing on patients after successful treatment. The information is organized and clear. When looking at prior studies in the field, i think it would be good to include/cite my colleagues' recent work published in The Oncologist, http://theoncologist.alphamedpress.org/content/early/2018/12/27/theoncologist.2018-0266.long, categorizing and providing suggested interventions for various types of weight loss that impact pancreatic cancer patients. There are a few very minor editorial mistakes, including the references cited as 1_3 instead of 1-3, and a few inconsistencies with capitalization (e.g., Upper GI Cancers on the lower half of page 2) and hyphenation (e.g., L-cells vs. L cells). 

Author Response

Dear Reviewer,

 Many thanks for your invaluable comments. We found your colleagues' recent paper very useful and relevant so we've cited this impressive article, as well (No. 82 in the References). 

 Moreover, I revised all minor editorial mistakes as requested. 

 Thank you again for your time and consideration.